



# A comprehensive history of climate and habitat stability of the last 800,000 years

Mario Krapp[1], Robert Beyer[1], Stephen L. Edmundson[1,2], Paul J. Valdes[3], and Andrea Manica[1]

[1]Department of Zoology, University of Cambridge, Downing Street, Cambridge CB2 3EJ, United Kingdom
[2]Department of Earth Sciences, Utrecht University, Budapestlaan 4, 3584 CD Utrecht, The Netherlands
[3]School of Geographical Sciences, University of Bristol, BS8 1SS Bristol, United Kingdom

**Correspondence:** Mario Krapp (mariokrapp@gmail.com)

**Abstract.** A detailed and accurate reconstruction of the past climate is essential in understanding the interactions between ecosystems and their environment through time. We know that climatic drivers have shaped the distribution and evolution of species, including our own, and their habitats. Yet, spatially-detailed climate reconstructions that continuously cover the Quaternary do not exist. This is mainly because no paleoclimate model can reconstruct regional-scale dynamics over geolog-
ical time scales. Here we develop a statistical emulator, the Global Climate Model Emulator (GCMET), which reconstructs the climate of the last 800,000 years with unprecedented spatial detail. GCMET captures the temporal dynamics of glacial-interglacial climates as an Earth System Model of Intermediate Complexity would whilst resolving the local dynamics with the accuracy of a Global Climate Model. It provides a new, unique resource to explore the climate of the Quaternary, which we use to investigate the long-term stability of major habitat types. We identify a number of stable pockets of habitat that have
remained unchanged over the last 800 thousand years, acting as potential long-term evolutionary refugia. Thus, the highly detailed, comprehensive overview of climatic changes through time delivered by GCMET provides the needed resolution to quantify the role of long term habitat change and fragmentation in an ecological and anthropological context.

## 1 Introduction

Current patterns of diversification within and between species, such as our own (Scerri et al., 2018), and the structuring of
whole ecosystems can only be studied in the context of past climatic changes that have shaped them through time (Doebeli and Dieckmann, 2003). A detailed understanding of such processes has become an urgent necessity in order to predict responses to global change. However, whilst predictions of climate change and their impacts over the next few tens or hundreds of years are based on comprehensive Global Climate Models (GCMs) that resolve processes at high temporal and spatial resolution, such as those used in the latest IPCC Assessment Report (Solomon et al., 2007), reconstructions back in time are challenging
as they have to span a much longer period. GCMs can provide snapshots for a specific time or short transients in the order of a few thousands of years, whilst periods of tens or hundreds of thousands of years can only be covered with Earth System Models of Intermediate Complexity (EMICs) (Ganopolski and Calov, 2011; Timmermann et al., 2013), at the cost of low spatial resolution and a simplified representation of the climate system (Claussen et al., 2002). Neither of those two types of





models is intentionally designed for paleo-ecology or species evolution, disciplines that require appropriate temporal scales of
up to hundreds of thousands of years and spatial scales down to tens of kilometres.

Here, we fill this gap for a long-term reconstruction of climate that resolves regional-scale dynamics by reconstructing the
last 800 thousand years (ka) at an unprecedented spatial resolution of approximately 1°. Unlike previous emulator approaches
which aimed at capturing climate dynamics at a global level (Araya-Melo et al., 2015; Lord et al., 2017), we explicitly focus
on the local emulation of climate. We critically evaluate our reconstructed 800 ka of climate history against proxy records,
demonstrating that our Global Climate Model Emulator (GCMET) provides a high quality reconstruction, equivalent to that
provided by global climate model simulations, for the last 800 ka.

The aim of this paper is twofold: 1) We present the technical details of our emulator approach (Sect. 2), which, in this
setup, can be thought of as an extension of the HadCM3 snapshot simulations into the deeper past. We validate GCMET (as an
emulator of HadCM3) with longer-term Quaternary climate proxies (Sect. 3). 2) We regard the resulting reconstructed climate
history of the last 800 ka as a high-value data set relevant for a wide range of applications in research areas that deal with
long-term past climate changes. As a case study, we reconstructed ecosystems throughout the last 800 ka (Sect. 4) and analysed
the long-term stability of human habitats through time in terms of their spatial fragmentation.

## 2 Methods and data

Our emulator approach consists of two steps (Fig. 1): a reconstruction of the global climate at moderate spatial resolution
followed by a more detailed representation of local dynamics using multiple snapshot simulations from the family of HadCM3
climate models (Valdes et al., 2017). In the first step, we use 72 simulations covering the past 120 ka from the HadCM3 cli-
mate model (Singarayer and Valdes, 2010; Davies-Barnard et al., 2017), with a resolution of 3° (2.5° × 3.75°), and build a
set of linear regression models that are the basis of GCMET. In the second step, we increase the resolution of our GCMET
reconstructions to about 1° (1.25°×0.83°) using high resolution HadAM3H (Hadley Centre Atmospheric Model 3, High reso-
lution) simulations covering the period of the last deglaciation. To do so we computed high-resolution difference maps between
equivalent HadAM3H and GCMET simulations.

### 2.1 HadCM3 and HadAM3H

HadCM3 is a fully coupled global climate model with an atmospheric component, HadAM3, which has a horizontal resolution
of 2.5° × 3.75°, with 19 vertical levels, and a time step of 30 min. The ocean and sea-ice component of HadCM3 has a
horizontal resolution of 1.25° × 1.25° and 20 vertical levels. In this paper, we use 72 available HadCM3 simulations covering
the last 120,000 years in 2,000-year intervals from 120,000 to 24,000 ka before present (BP) and in 1,000-year intervals from
22,000 to present-day (Singarayer and Valdes, 2010; Davies-Barnard et al., 2017) (https://www.paleo.bristol.ac.uk/ummodel/
data/tdwza/standard_html/tdwza.html, last accessed on 05 Oct 2018).

HadAM3H is the higher resolution version of the atmosphere model HadAM3. It has a horizontal resolution of 1.25° by
0.83° with 30 vertical levels and a time step of 10 min. HadAM3H uses the surface fluxes from the associated fully coupled





HadCM3 simulations. HadAM3H is available for the time since the last deglaciation, 21 ka BP, i.e., 21, 18, 15, 12, 10, 8, 6, 3, and 0 ka BP.

## 2.2 The global climate model emulator GCMET

GCMET relies on a set of linear regression models, one for each individual time series of every HadCM3 model grid box

with the following independent variables: atmospheric $CO_2$ concentrations (as a major greenhouse gas), and three variables reflecting the orbital forcing (Berger and Loutre, 1991). These are based on obliquity $\varepsilon$ and two combinations of eccentricity $e$ and precession $\omega$: $e \sin \omega$, henceforth referred to as precession index I, and $e \cos \omega$ (precession index II). These are a generally accepted set of orbital forcings as they reflect insolation at any location and any time (Araya-Melo et al., 2015; Lord et al., 2017). Atmospheric $CO_2$ concentrations are the same as in the respective HadCM3 time slice simulation, e.g., 280 ppmv for

0 ka BP.

Our approach interpolates the HadCM3 model output at each grid box, i.e., a time series, to the parameter settings for which HadCM3 has been evaluated, i.e., the time series of $CO_2$ concentrations and the three orbital parameters. This is unlike state-of-the-art emulators that extrapolate to settings which have not been evaluated. However, it ensures that the predictions by GCMET are well behaved. As a consequence, GCMET is only valid within the boundary conditions for which it has been

parameterised; in our case, we build valid regression coefficients for the Quaternary. These limitation could be easily alleviated by any additional snapshot simulation with a new, expanded parameter setting.

The dependent variables, i.e, the predictands, are climate variables such as temperature $T$, precipitation $P$, or specific humidity $Q$. The independent variables, i.e., the predictors, are applied as normalised forcings. Thus, the resulting regression coefficients, denoted as $\beta$ coefficients, can be compared across different climate variables, i.e., temperature and precipitation,

and across each other (Fig. 2).

Let us assume $Y(x,y,t)$ is a time series of a climate variable at a specific location $(x,y)$ at time $t$. To explain variations of $Y$ around a mean value $\overline{Y}$, i.e, $Y' = Y - \overline{Y}$, we run a multiple linear regression model for the anomalies $Y'$:

$$Y'(x,y,t) = \underbrace{\beta_\varepsilon(x,y)\varepsilon'(t) + \beta_{e\cos\omega}(x,y)(e\cos\omega)'(t) + \beta_{e\cos\omega}(x,y)(e\cos\omega)'(t)}_{\text{orbital forcing}}$$

$$+ \underbrace{\beta_{CO_2}(x,y)CO_2'(t)}_{\text{greenhouse gas forcing}} + \underbrace{\beta_M(x,y)M(x,y,t)}_{\text{variable land–sea mask effect}} \tag{1}$$

Here, the $\beta$s are the regression coefficients for the respective predictor (see Fig. 2 for maps of $\beta$ coefficients). We also consider changes in surface type, i.e., ocean (0), land (1), and ice (2) which have been masked as $M(x,y,t) \in [0,1,2]$. For example, around coastlines, land grid boxes can turn into ocean grid boxes when sea level is high. Similarly, expanding ice sheets turn land grid boxes into ice-covered grid boxes, and the climate variable $Y$ may respond to different surface types in different ways. To make the linear regression statistically well-conditioned, all independent variables have been normalised, i.e.,

the mean has been subtracted and the data has then been divided by their standard deviation. To prevent our linear regression



model from predicting negative precipitation values, we apply a logarithmic transformation which maps values from $[0,+\infty]$ to $[-\infty,+\infty]$. For bounded variables such as precipitation this is a common procedure. In the case of precipitation, the linear regression coefficients predict the response in terms of anomalies in the exponent.

The units of specific humidity are [kg/kg] and its values fall in the range between 0 and 1. For this reason, we transform specific humidity using the logit function, $\text{logit}(x) = \log(\frac{x}{1-x})$, which maps values from [0,1] to $[-\infty,+\infty]$. The decomposition of temperature $T$, precipitation $P$, and specific humidity $Q$ into anomalies, i.e., the $Y'$ on the left hand side of Eq. 1 is:

$$T = \overline{T} + \underbrace{T'}_{\hat{=}Y'} \tag{2}$$

$$\log(P) = \overline{\log(P)} + \underbrace{\log(P)'}_{\hat{=}Y'} \tag{3}$$

$$\text{logit}(Q) = \overline{\text{logit}(Q)} + \underbrace{\text{logit}(Q)'}_{\hat{=}Y'} \tag{4}$$

In contrast to existing emulator approaches (Araya-Melo et al., 2015; Lord et al., 2017; Rangel et al., 2018), our reconstructions are local-scale reconstructions which are in reasonable agreement with existing paleo-climate proxies as shown in our comprehensive model–data comparison below. Furthermore, because the parameter sampling is based on realistic glacial cycle snapshot simulations, the obtained regression coefficients are good enough approximations to predict previous Quaternary glacial–interglacial climate states well.

### 2.2.1 Training and test data

In order to make useful predictions and to evaluate the skill of our model, we divide the HadCM3 snapshots into and training and a test data set. A sensible choice is to use 80% of the HadCM3 snapshots to train the linear regression model and 20% to test it. For a 80/20 division of the 72 time slices into training and test data, i.e., 14 or 58 out of 72, there are $\binom{n}{k} = \binom{72}{58} \approx 3 \times 10^{14}$ possible combinations. But instead of randomly dividing the data into the training/test data, we follow an approach with the aim to preserve as much variance as possible in the training data, i.e. maximise the variance of the predictors. This is best illustrated by the phase plots of the predictors (Fig. 3). The training data set covers the edges of each phase plane and thus maximises the phase space covered by the linear regression model. This choice of training data ensures that the linear regression model interpolates within the phase space and does not need to extrapolate for the test data.

The procedure is as follows. We calculate the covariance matrix of the full parameter set ($n$=72) and its eigenvalues. Then, we randomly create a training data set ($k$=58) for which we compute the covariance. If the covariance of this sample training set is larger than the full covariance matrix, i.e., the eigenvalues of the covariance matrix are larger than the eigenvalues of the covariance matrix of the full parameter set, this sample parameter set is marked as a candidate for the final training set. After several iterations (N=10,000), we sum up how many times each time slice has appeared within a candidate training set. We then rank all time slices according to this number. In the final step, we pick the 80% top-ranked time slices as training data.



### 2.2.2 Validation

For the validation of GCMET, we use the proportion of variance in the response explained by the regression ($R^2$, a goodness of fit estimator of the training data; Fig. 4), and the root mean squared error (RMSE), an estimator of the goodness of the model for the prediction of the test data. Overall, our linear model is a better predictor for temperature than for precipitation.

Temperature responds more directly to local forcings because it is determined by the energy balance of downward and upward longwave and shortwave radiation and turbulent heat fluxes. The downward shortwave radiation depends on incoming solar radiation that is determined by orbital variations, whereas downward longwave radiation is determined by greenhouse gases such as $CO_2$ and water vapour, as well as cloud cover. Large-scale atmospheric circulation changes have a much smaller effect on temperature. Locally, it is therefore locally far better constrained by global $CO_2$ and orbital variations. This increases the predictive skill of our linear regression model substantially resulting in higher $R^2$ values and lower RMSEs.

The matter is more complicated for precipitation because it is a product of the hydrological cycle, which itself depends mainly on large-scale atmospheric dynamics, such as monsoonal systems in the tropics and subtropics, or mid-latitude storm systems. Local interactions between the atmosphere and the surface, such as evaporation and transpiration over the ocean, or deep convection over the tropics, matter to a lesser extent. Processes and circulation features like moisture transport or the atmospheric Hadley cell dynamics determine the non-local response of precipitation to $CO_2$ or orbital variations to a much larger extent. Because of the larger dynamical component of the hydrological cycle, compared to temperature, precipitation is much less constrained by external forcings than temperature. Thus, the linear regression model has less predictive skill for precipitation than for temperature. However, we find that the predictive skill for precipitation can be improved by using temperature and specific humidity as predictors instead of the orbital parameters and $CO_2$. In this way, the RMSEs can be substantially reduced, especially over land (Fig. 4E).

For the improved precipitation model we used temperature $T$ and specific humidity $Q$ as independent variables, i.e.,

$$\log(P)' = \beta_T T' + \beta_Q \mathrm{logit}(Q)'. \tag{5}$$

For the prediction of climate before $120\,\mathrm{ka\,BP}$ this implies that we first need to reconstruct $T$ and $Q$, and then use the obtained $\beta$ coefficients for $T$ and $Q$ to reconstruct $P$.

### 2.2.3 The regression coefficients

To assess how reliable our predictor estimates are, we calculate the p-values for each of the predictors, i.e., the significance of our $\beta$ coefficients. Here, the p-value tests the null hypothesis whether the coefficient is equal to zero, i.e., the specific predictor has no effect. If the p-value is below a certain threshold—in our case below the 5% significance level: $p < 0.05$—the null hypothesis can be rejected. That means that the specific predictor is a meaningful addition to our linear regression model and any changes in the associated predictor are related to changes in the corresponding climate variable. Regions for which the null hypothesis cannot be rejected are displayed as shaded and hatched in Fig. 2.



## 2.3 Dynamic downscaling to higher resolution

Using nine high-resolution HadAM3H simulations covering the deglaciation since 21 ka BP (21, 18, 15, 12, 10, 8, 6, 3, and 0 ka BP), we are able to increase the spatial resolution from 3°, which is the spatial resolution of GCMET after the linear regression step (and the same as the coarse resolution of the original HadCM3 snapshots), to ca. 1°. We do so by calculating

the difference between equivalent coarse- and high-resolution snapshots. For example, the difference at 10 ka BP is $\Delta_{10\,ka\,BP} = HadAM3H_{10\,ka\,BP} - HadCM3_{10\,ka\,BP}$. We choose to interpolate the differences linearly according to their $CO_2$ levels, e.g., 231 ppm at 10 ka BP, because any statistical model with more than one variable would require more snapshots to adequately predict the differences. Thus, we simply assume that the differences between a coarse- and high-resolution climate can be explained as a function of the $CO_2$ forcing, i.e., $\Delta_{10\,ka\,BP} = \Delta_{231\,ppm}$. For any period in the past, e.g., 300 ka BP, we add the

high-resolution difference, i.e., the $\Delta$, which corresponds to the respective $CO_2$ level, to the coarse-resolution reconstruction. Note that the downscaling approach captures the regional-scale dynamics of the GCM in this step, which change over time. This is in contrast to the commonly used "delta method" for downscaling of climate model data which assumes a constant difference between simulated and observed data.

To illustrate the importance of higher spatial variability, we compared the high-resolution version of GCMET, the origi-

nal resolution version denoted as GCMET-LO, and LOVECLIM, an EMIC with a horizontal resolution of ca 5.5°×5.5°, to present-day observations (ERA-20C re-analysis 1961–1990 average (Poli et al., 2016)). Due to their lower spatial resolution LOVECLIM and GCMET-LO cannot capture the observed continental climate patterns, whereas GCMET resolves those spatial features well (Fig. 5B).

## 2.4 Boundary conditions: CO₂, global sea-level, and Northern Hemisphere ice sheets

For realistic high-resolution reconstructions the model boundary conditions for the last 800 ka need to be known: atmospheric $CO_2$ levels and orbital parameters (Fig. 3A), global sea level (for the land-sea mask), and the extent of Northern Hemisphere (NH) ice sheets. The longest, quasi-continuous record of past $CO_2$ levels is the 800,000 years long $CO_2$ record from the EPICA Dome C ice core in Antarctica (Bereiter et al., 2015). The orbital parameters are from the same data set as before (Berger and Loutre, 1991). For ice sheet extents, we use model outputs from CLIMBER-2/SICOPOLIS simulations (Ganopolski and Calov,

2011) for which NH ice sheet extents and heights are available for the last 800 ka in 1 ka-year intervals, from which we use the period from 800–123 ka BP . For 122–0 ka BP, we use the ice sheet configurations from the ICE-6G data set (Peltier et al., 2014) (http://www.atmosp.physics.utoronto.ca/~peltier/data.php, last accessed 09.11.2018). Changes in the coast lines affecting the land–sea mask are derived from a global sea-level record (Spratt and Lisiecki, 2016) which has been added on top of present-day coast lines while preserving inland lakes.



## 3  Comparing GCMET to climate proxies


Despite the increasing number of available paleoclimate proxies, only a small fraction can be used for a quantitative comparison to climate models as translating sediment core data into actual climate variables remains a difficult task. Marine sediment cores are valuable archives of past sea surface temperature (SST) records. Because their associated biogeochemistry is relatively straightforward, marine proxies can be utilised as so-called paleo-thermometers and are thus well suited for a direct

proxy–model comparison. For these proxies, we make a direct comparison between mean annual temperature (MAT) and SST, quantified both in terms of correlation between the predicted and observed time series and the RMSE. We note that MAT and SST are not the same climatological quantities; SST is the temperature of the ocean surface and has a lower limit of about -1.8°C, the freezing point of saltwater. While we expect MAT and SST to co-vary in low and mid-latitudes, at higher latitudes, seasonal or perennial sea ice cover could make a comparison between both variables problematic.

For terrestrial proxies, for which a translation into climatic variables is not straightforward, we simply quantify the correlation between the two time series. However, the interpretation of terrestrial proxies from a climate perspective can also be problematic. For example, pollen-based vegetation reconstructions are suggested to be less reliable as climate proxies, particularly for interglacials (Herzschuh et al., 2016). Other land-based proxies such as dust deposits integrate long-term climatic changes over large regions and hence do not necessarily capture climatic effects at their specific location.

For the comparison of GCMET against proxy reconstruction, we assembled long-term marine SST and terrestrial climate proxy reconstructions (Figs. 7,8) which cover a period of at least 150 ka BP during the last 800 ka (Tables 1 and 2). We assessed the goodness of the long-term GCMET climate reconstructions by cross-comparing those SST proxy reconstructions to the MAT reconstruction by GCMET and to surface temperature data from the LOVECLIM climate model (Timmermann et al., 2013) (http://apdrc.soest.hawaii.edu/projects/paleomodeling/800k.php, last accessed 08/11/2018).

Reconstructions by GCMET are in good agreement with a number of marine records (Fig. 7B-E), with a mean RMSE of 1.5 K for all SST proxies and a mean correlation of 0.5. GCMET captures long-term temporal dynamics of glacial–interglacial climates and its performance in that respect is on average as good as LOVECLIM's (r=0.5, Fig. 5B). Despite the diverse nature of the terrestrial proxies (e.g. speleothems, loess, pollen), GCMET performance was as good as for marine proxies (r=0.5, Fig. 5B and Fig. 8B,C).

## 4  Ecosystem reconstructions and habitat stability over the last 800 ka


In the final part of this paper, we highlight the potential of a comprehensive, long-term climate data set. As an example we investigate ecosystem stability (Fig. 9) over the last 800 ka, focusing on 14 major terrestrial habitats defined by the WWF *Global 200* (Olson et al., 2001) (Fig. 9A). The motivation for this is that habitat fragmentation, which is closely related to ecosystem (in)stabilities, may have played a key role in the evolution of our species, *homo sapiens* (Scerri et al., 2018).

We use a random forest classifier (Breiman, 2001) that is trained on a set of four climate variables from GCMET (minimum and mean annual temperature, and minimum and mean annual precipitation) to reconstruct the present-day distribution of the 14 ecoregions. The required present-day data was split into a training (80%) and a test data set (20%). The classification factors,



i.e., the analogue of predictors in a linear model, from this training data set were then applied to predict ecosystem changes of the last 800,000 years.

The goodness of the predictions by the random forest classifier can be estimated by the so-called receiver operating characteristic (ROC, Fig. 9D). A ROC curve displays the true positive rate against the false positive rate and the closer that curve is to the upper left corner, the better the prediction for a specific ecosystem is. For example, the point at coordinate (0,1) represents the best possible prediction with 100% sensitivity (i.e., no false negatives) and 100% specificity (i.e., no false positives). The diagonal line corresponds to a prediction by random guessing.

The random forest classification is very close to perfect classification for the average of all ecosystem types and the area under the curve is an estimator for the goodness of the classification (numbers are given in legend of Fig. 9). Except for a few instances, such as for "Tropical & Subtropical Coniferous Forests" and "Mangroves", values are always larger than 0.9 (average 0.98).

The spatially detailed reconstructions provided by GCMET allow us to explore the effect of climate on habitats over time. As can be seen in Fig. 9C, stability depends on location, with sparsely vegetated regions such as deserts among the most stable habitats in the world, the others being the core tropical rainforests along the equator. Large parts of Eurasia and North America are rendered unstable by the advancing and retreating NH ice sheets with ecosystems alternating between vast forests during the warm interglacials and large tundras during the cold glacials (an animated version of the habitat changes throughout the last 800 ka is available as *Supplementary Video*). However, a few fragmented core boreal forest habitats remain.

## 5 Summary and discussion

The global climate model emulator, GCMET, is an effective and computationally efficient tool to reconstruct the climate of the past over long periods while based on only a relatively small number of paleoclimate simulations. In contrast to other emulators, our grid box by grid box approach assumes that the spatial correlations are sufficiently well preserved by HadCM3 and thus are well represented in the GCMET reconstructions as can be seen in the $\beta$ coefficient maps in Fig. 2. The glacial and interglacial climate states of the Quaternary with its respective glacial and interglacial boundary conditions are an ideal testbed to show how well GCMET emulates the HadCM3 climate response without explicitly running any HadCM3 simulation for earlier times. GCMET can in principle be applied to any other global climate model for which a reasonable number of simulations exists.

Our approach has several limitations. For example, we are unable to account for climate variability on time scales that are not simulated in the underlying HadCM3 snapshots. Originally, these have been set up to examine climate changes on orbital timescales and they are not fully transient (Singarayer and Valdes, 2010). This has implications for the ecosystem reconstructions and the inferred habitat stability because we can only account for climate and ecosystem variability on time scales longer than 1000 years and more. However, the assumption of a equilibrium climate in GCMET is not too far fetched given the relatively good level of agreement with climate proxies, most of which have a temporal resolution of more than 1000 years, and as such can be thought of as reflecting the average long-term rather than an instantaneous climatic state.



Another limitation is that we cannot link the regression coefficients for $CO_2$ and the orbital parameters to true geophysical processes. GCMET is a purely statistical model and as such any physical interpretation of $\beta$ coefficients would be flawed. However, the imposed $CO_2$ forcing from the Antarctic ice core with its distinct late Quaternary, 100,000 year cyclicity already accounts for the orbital effects on the Earth system. A different but interesting question would be whether it is possible to disentangle orbital changes from the natural variations in $CO_2$; however, this is beyond the scope of this paper.

Because GCMET is a statistical emulator, we can use it to assess the uncertainty of the predictors (i.e., boundary conditions) and their respective coefficient estimates. We did this for the global mean temperature comparison (Fig. 5A) but the uncertainty in the $CO_2$ forcing is fairly small as are the standard errors for the temperature regression (not shown), and thus, the uncertainty is hardly visible (blue shaded). In this specific example we also see how much GCMET depends on the underlying model performance. GCMET temperature reconstructions closely follow the HadCM3 curve for the last 120 ka simulations but differ

from the proxy record during certain periods of the past. For example, HadCM3 underestimates the global mean temperature 120 ka ago compared to the proxy record and so does GCMET. For earlier periods, GCMET can be used to detect discrepancies between the GCM and long-term paleo records as we did in the model–proxy comparison in Sect. 3. The performance of an emulator strongly depends on the representation of the climate system in the underlying GCM: Realistic climate reconstructions rely on both a realistic climate model and a good statistical model, i.e., emulator.

## 255   6   Conclusions

A major advantage of GCMET is that it is computationally inexpensive. GCMET can produce high-quality reconstructions of the last 800 ka that compare well with proxy records of the past. This is the equivalent of hundreds of GCM snapshots, a prohibitive endeavour for the foreseeable future. A way to understand the fit of GCMET predictions against climate proxy time series is that our approach captures the quasi-equilibrium state of the climate system, thus allowing us to efficiently describe

the behaviour over the longer, millennial, time scales. In turns, this implies that the glacial-interglacial climate of the Middle and Late Pleistocene responded in a consistent manner to orbital forcings and $CO_2$. It will be interesting to test whether this approximation is also valid for the Early Pleistocene with its faster ice age cyclicity of 41 ka; for this endeavour, we currently lack quasi-continuous $CO_2$ estimates before the Mid-Pleistocene Transition, however GCMET is fully capable of covering the appropriate time periods if estimates became available. For the moment, we can offer a detailed, coherent reconstruction of the

past 800 thousand years, which allowed us to pinpoint long-term potential refugia that have been characterised by the same habitat, and we expect that this will open up new ways to study the impact of past climate in a number of disciplines such as ecology and anthropology.

*Code and data availability.*   The generated climate reconstructions for the last 800,000 years are publicly available in the project's *Open Science Framework* repository at this address: https://bit.ly/2XWrGvF. This data set comes in two version, one is the output after the dynamical

downscaling step and one that has been bias corrected afterwards using the ERA-20C climatology from 1961–1990 (Poli et al., 2016). The



variables listed in Table 3 are available in 1000 year intervals and in 1°, the HadAM3H resolution, and 1.125° horizontal, i.e, the ERA-20C grid resolution under the project's `data` directory. The individual proxy time series and their respective GCMET counterpart as shown in Fig. 7 and 8 are available as Excel spreadsheet (`output_150ka.xlsx`) in the same directory.

Model code for GCMET as well as the code for the analysis and visualisation of figures is also publicly available at the same address
under the project's `code` directory (`gcmet.tar.gz` and required input files `inputs.tar.gz`). NetCDF files have been processed using `cdo` (Schulzweida, 2019). The linear regression for GCMET is based on the Python library `statsmodels` (Seabold and Perktold, 2010). The random forest classifier has been implemented using the Python library `scikit-learn` (Pedregosa et al., 2011). All visualisations are made with `matplotlib` (Hunter, 2007) and `cartopy` (Met Office, 2010).

*Video supplement.* Movies of the habitat evolution over the last 800,000 years can be found in the *Open Science Framework* repository
(https://bit.ly/2XWrGvF) under the project's `movies` directory. The 14 major WWF habitats (`ecosystems_800ka.mp4`) have been aggregated into the four categories: snow and ice, barren and sparsely vegetated, open habitat, and forests (`habitats_800ka.mp4`). The temporal evolution of the atmospheric $CO_2$ concentrations are shown below the map.

*Author contributions.* AM and MK devised the project. MK devised and implemented the emulator with input from RB and SLE. PJV provided additional *HadCM3* snapshot simulations. MK and AM wrote a first draft of the paper which was improved by input from all other
authors. MK prepared the figures and videos.

*Competing interests.* The authors declare no conflict of interest.

*Acknowledgements.* This work was supported by an ERC Consolidator Grant to AM (Local Adaptation 647787). MK wishes to thank Eric Wolff and Max Holloway for commenting on an earlier version of this manuscript.



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



**Table 1.** Marine proxy records that have been used in the validation of the climate reconstruction, their coordinates, correlation coefficients, types, and respective references.

| core/name | lon | lat | corr coeff | type | reference(s) |
|---|---|---|---|---|---|
| DSDP 594 | 175.0 | -45.5 | 0.60 | SST | PIGS, Schaefer et al. 2005 |
| DSDP 607 | -33.0 | 41.0 | 0.46 | SST | PIGS, Ruddiman et al. 1989 |
| GeoB 1105 | -12.4 | -1.7 | 0.62 | SST | Nürnberg et al. 2000 |
| GeoB 1112 | -10.7 | -5.8 | 0.53 | SST | Nürnberg et al. 2000 |
| HY04 | -95.0 | 4.0 | 0.30 | SST | PIGS, Horikawa et al. 2010 |
| MD01-2444 | -10.1 | 37.6 | 0.71 | SST | Martrat et al. 2007 |
| MD02-2529 | -84.1 | 8.2 | 0.53 | SST | Rincón-Martínez and Leduc 2012 |
| MD03-2699 | -10.7 | 39.0 | 0.61 | SST | Rodrigues et al. 2011 |
| MD06-2986 | 167.9 | -43.4 | 0.70 | SST | PIGS, Hayward et al. 2012 |
| MD06-3018 | 166.2 | -22.6 | 0.38 | SST | Russon et al. 2010 |
| MD85-668 | 46.0 | 0.0 | 0.48 | SST | Bard et al. 1997 |
| MD90-963 | 73.9 | 5.1 | 0.50 | SST | Rostek et al. 1993 |
| MD96-2048 | 36.0 | -26.2 | 0.52 | SST | Caley et al. 2011 |
| MD97-2120 | 174.9 | -45.5 | 0.73 | SST | Pahnke et al. 2003 |
| MD97-2140 | 141.5 | 2.0 | 0.55 | SST | PIGS, Garidel-Thoron et al. 2005 |
| ODP 1012 | -118.4 | 32.3 | 0.59 | SST | PIGS, Liu et al. 2005 |
| ODP 1014 | -118.9 | 32.8 | 0.75 | SST | Yamamoto et al. 2007 |
| ODP 1020 | -126.4 | 41.0 | 0.50 | SST | PIGS, Herbert 2001 |
| ODP 1077b | 10.4 | -5.2 | 0.18 | SST | Schefuß et al. 2004 |
| ODP 1082 | 11.8 | -21.1 | 0.43 | SST | Etourneau et al. 2009 |
| ODP 1087 | 15.3 | -31.5 | 0.13 | SST | McClymont et al. 2005 |
| ODP 1090 | 8.9 | -42.9 | 0.69 | SST | PIGS, Martínez-Garcia et al. 2009 |
| ODP 1123 | -171.5 | -41.8 | 0.35 | SST | PIGS, Crundwell et al. 2008 |
| ODP 1125 | -178.2 | -42.6 | 0.55 | SST | Hayward et al. 2008 |
| ODP 1143 | 113.3 | 9.4 | 0.61 | SST | PIGS, Li et al. 2011 |
| ODP 1146 | 116.3 | 19.5 | 0.53 | SST | PIGS, Herbert et al. 2010 |
| ODP 1172 | 149.9 | -44.0 | 0.28 | SST | Nürnberg and Groeneveld 2006 |
| ODP 1239 | -82.1 | -0.7 | 0.53 | SST | Dyez et al. 2016 |
| ODP 306 | -27.9 | 56.4 | 0.33 | SST | Alonso-Garcia et al. 2011 |
| ODP 722 | 59.8 | 16.6 | 0.45 | SST | PIGS, Herbert et al. 2010 |
| ODP 806b | 159.4 | 0.3 | 0.55 | SST | PIGS, Medina-Elizalde and W Lea 2005 |
| ODP 846 | -90.8 | -3.1 | 0.49 | SST | PIGS, Liu 2004 |
| ODP 871 | 172.3 | 5.6 | 0.63 | SST | Dyez and Ravelo 2013 |
| ODP 882 | 167.6 | 50.4 | 0.11 | SST | Martínez-Garcia et al. 2010 |
| ODP 977A | 0.0 | 37.5 | 0.63 | SST | Martrat et al. 2007 |
| ODP 982 | -15.9 | 57.5 | 0.35 | SST | PIGS, Lawrence et al. 2009 |
| ODP 999 | -78.7 | 12.8 | 0.21 | SST | Schmidt et al. 2006 |
| PS75034-2 | -80.1 | -54.4 | 0.79 | SST | PIGS, Ho et al. 2012 |
| RC09-166 | 48.8 | 12.5 | 0.35 | SST | Tierney et al. 2017 |

PIGS refers to Past Interglacials Working Group of PAGES (2016)



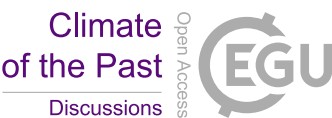

**Table 2.** Terrestrial proxy records that have been used in the validation of the climate reconstruction, their coordinates, correlation coefficients, types, and respective references.

| core/name | lon | lat | corr coeff | type | reference(s) |
|---|---|---|---|---|---|
| Baoji, China | 107.1 | 34.4 | 0.62 | rainfall | Beck et al. 2018 |
| Bittoo | 77.8 | 30.8 | -0.43 | $\delta^{18}$O | Kathayat et al. 2016 |
| Chanwu | 107.7 | 35.2 | 0.58 | $\delta^{18}$O | Guo et al. 2009 |
| Clearwater | 114.9 | 4.1 | -0.47 | $\delta^{18}$O | Carolin et al. 2016 |
| Dead Sea | 35.0 | 30.5 | -0.68 | lake level | Waldmann et al. 2010 |
| Devil's Hole | -116.3 | 36.4 | 0.67 | $\delta^{18}$O | Landwehr et al. 2011 |
| Duhlata | 23.2 | 42.5 | 0.41 | ODL | Stoykova et al. 2008 |
| EPICA Dome C | 123.4 | -75.0 | 0.91 | temperature | Jouzel et al. 2007 |
| Kesang | 81.8 | 42.9 | -0.15 | $\delta^{18}$O | Cheng et al. 2012 |
| Lake Baikal | 108.4 | 53.7 | -0.16 | Bio. sil. | Prokopenko et al. 2006 |
| Lake El'gygytgyn | 172.0 | 67.5 | 0.17 | mag. susc. | Melles et al. 2012 |
| Negev | 34.8 | 30.6 | -0.61 | $\delta^{18}$O | Vaks et al. 2010 |
| Peqiin | 36.0 | 32.6 | -0.67 | $\delta^{18}$O | Bar-Matthews et al. 2003 |
| Sanbao-Dongge | 110.4 | 31.7 | 0.12 | $\delta^{18}$O | Cheng et al. 2016 |
| Soreq | 36.0 | 31.4 | -0.56 | $\delta^{18}$O | Bar-Matthews et al. 2003 |
| Tenaghi Philippon | 24.2 | 41.0 | 0.65 | arb. pollen | PIGS, Tzedakis et al. 2006 |
| Tzavoa | 35.2 | 31.2 | -0.37 | $\delta^{18}$O | Vaks et al. 2006 |
| Weinan | 109.6 | 34.4 | 0.48 | temperature | K. Thomas et al. 2016 |
| Xifeng loess | 107.6 | 35.7 | 0.60 | Fed/Fet | Guo et al. 2009 |
| Yimaguan Luochuan | 108.5 | 35.8 | 0.60 | mag. susc. | PIGS, Hao et al. 2012 |

PIGS refers to Past Interglacials Working Group of PAGES (2016)





**Table 3.** List of file names for GCMET regression results and climate reconstructions of the last 800 ka as well as the Excel spreadsheet for the proxy–model comaprison. This data can be found in the *Open Science Framework* repository https://bit.ly/2XWrGvF under the project's `data` directory.

| | variable | file name |
|---|---|---|
| regression results | MAT | `hadcm3_000-120_ann_regression_temp_co2-ecospre-esinpre-obl.nc` |
| | MAP | `hadcm3_000-120_ann_regression_prec_co2-ecospre-esinpre-obl.nc` |
| | MAQ | `hadcm3_000-120_ann_regression_shum_co2-ecospre-esinpre-obl.nc` |
| | MAP (MAT and MAQ) | `hadcm3_000-120_ann_regression_prec_shum-temp.nc` |
| | MINT | `hadcm3_000-120_min_regression_temp_co2-ecospre-esinpre-obl.nc` |
| | MINQ | `hadcm3_000-120_min_regression_shum_co2-ecospre-esinpre-obl.nc` |
| | MINP (MINT and MINQ) | `hadcm3_000-120_min_regression_prec_shum-temp.nc` |
| reconstructions | MAT | `temp_800ka_ann_hi_nobc.nc` |
| | MINT | `temp_800ka_min_hi_nobc.nc` |
| | MAP | `prec_800ka_ann_hi_nobc.nc` |
| | MINP | `prec_800ka_min_hi_nobc.nc` |
| | WWF 14 major habitats | `ecosystems_random_forests_wwf_tmean_tmin_pmean_pmin.nc` |
| | MAT (BC) | `temp_800ka_ann_hi_20c_bc.nc` |
| | MINT (BC) | `temp_800ka_min_hi_20c_bc.nc` |
| | MAP (BC) | `prec_800ka_ann_hi_20c_bc.nc` |
| | MINP (BC) | `prec_800ka_min_hi_20c_bc.nc` |
| | Proxy comparison | `output_150ka.xlsx` |

MAT–mean annual temperature; MINT–minimum temperature; MAP–mean annual precipitation; MINP–minimum precipitation; MAQ–mean annual specific humidity; MINQ–minimum specific humidity

BC–bias correction with ERA-20C 1961–1990 climatology




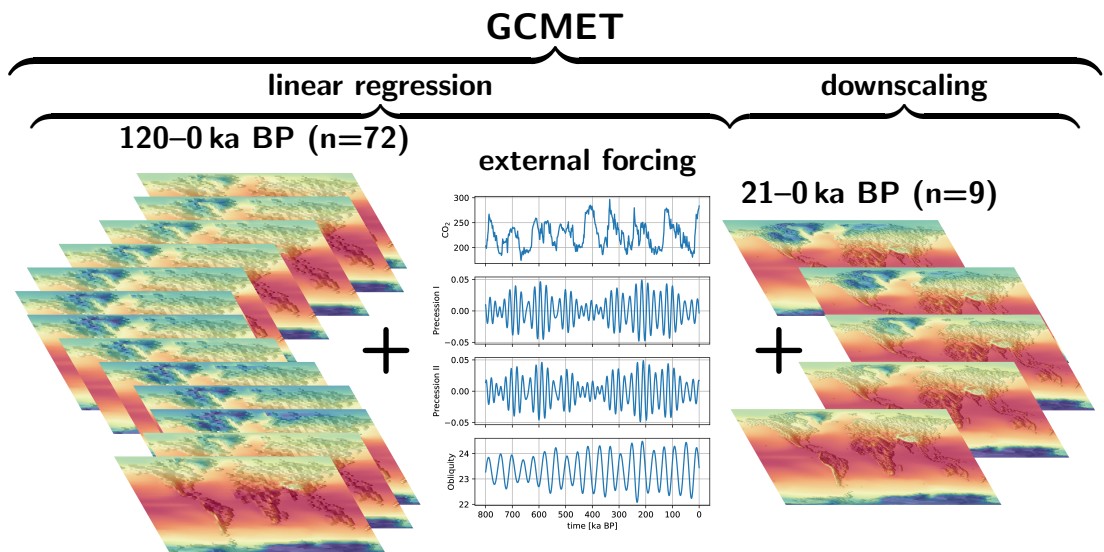

**Figure 1.** Schematic of the GCMET components: A linear regression combines 72 HadCM3 snapshot simulations with the external forcings, i.e., $CO_2$ and the three orbital parameters, which provides the basis of the long-term climate reconstructions of the last 800 thousand (or 2 million) years. Using 9 high-resolution snapshots covering the last deglaciation provides the basis of the downscaling approach based on $CO_2$ which yields the final high-resolution long-term climate reconstructions of GCMET.



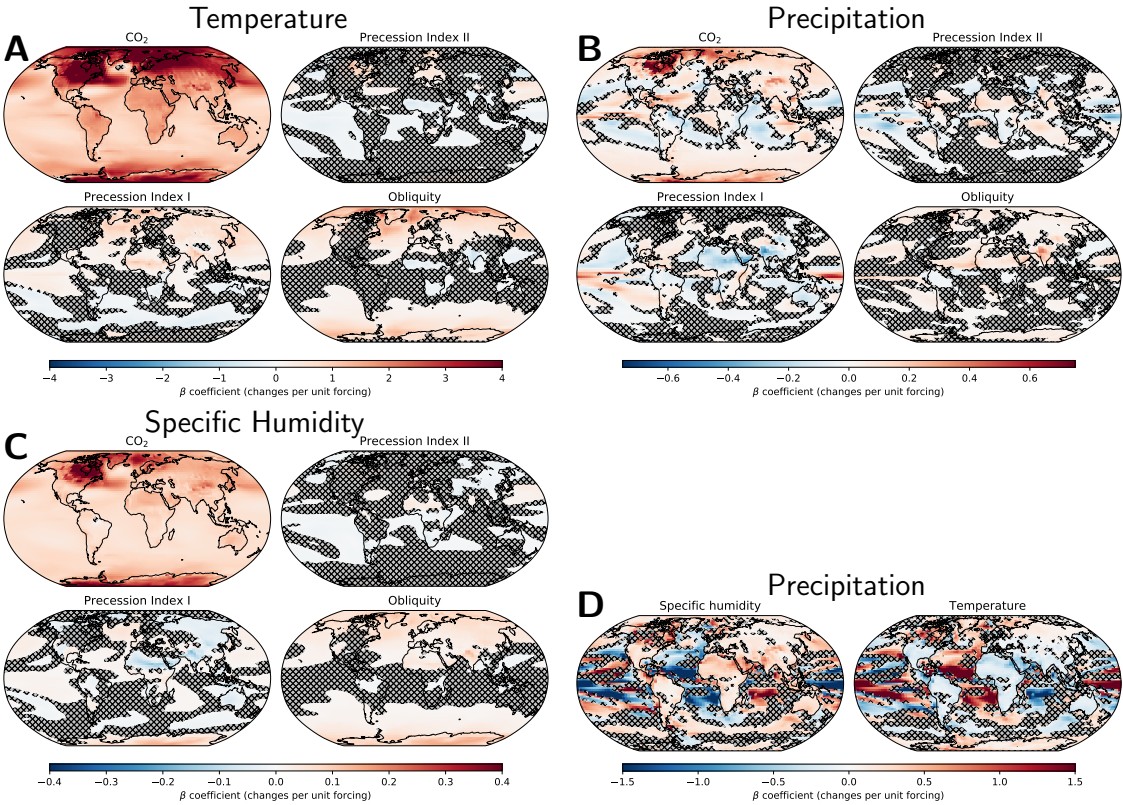

**Figure 2.** Regression coefficients, i.e., $\beta$ coefficients, for (A) mean annual temperature, (B) precipitation, (C), specific humidity, and (D) the alternative model for precipitation—based on temperature and specific humidity. Regions where the respective coefficient is not statistically significant ($p < 0.05$) are hatched and shaded.




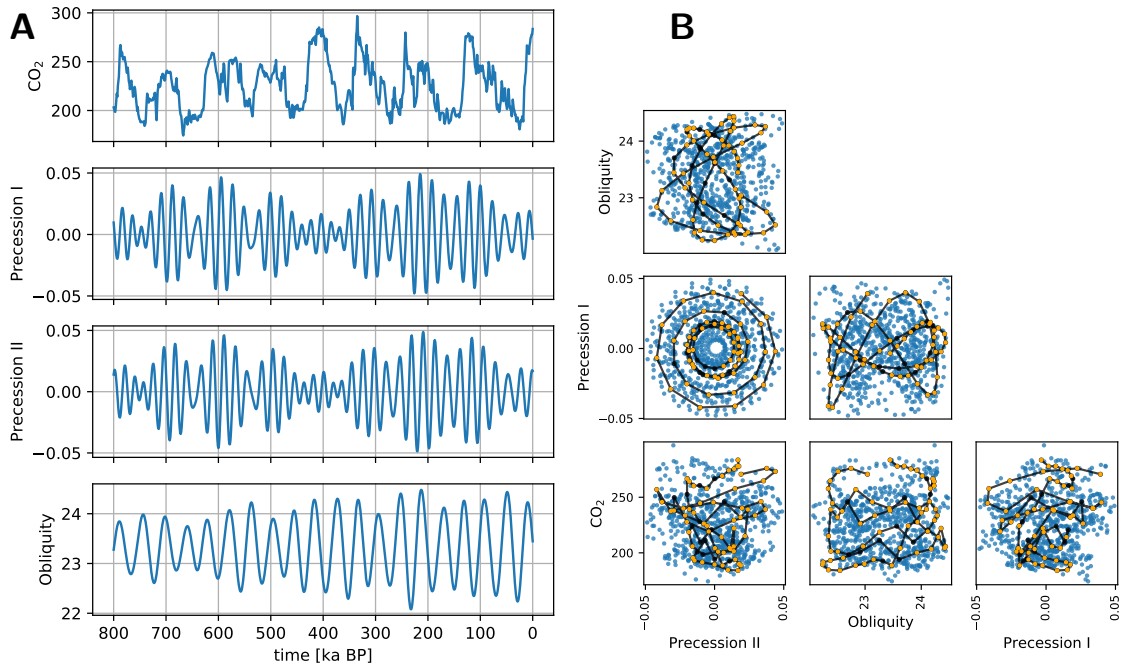

**Figure 3.** (A) Time series of the four external parameters: $CO_2$ and orbital parameters for the last 800 ka and (B) the associated parameter space as scatter plot matrix (blue dots). The continuous $CO_2$ record is from the EPICA Dome C ice core in Antarctica (Bereiter et al., 2015). The orbital parameters are numerical solutions for the Earth's orbit and rotation in terms of eccentricity, precession, and obliquity (Berger and Loutre, 1991). In (B), black lines with black dots represent the total 72 parameter sets. Orange dots highlight the parameter sets of the 58 HadCM3 snapshot simulations which we used as training data (80% of the total 72) for the linear regression model.

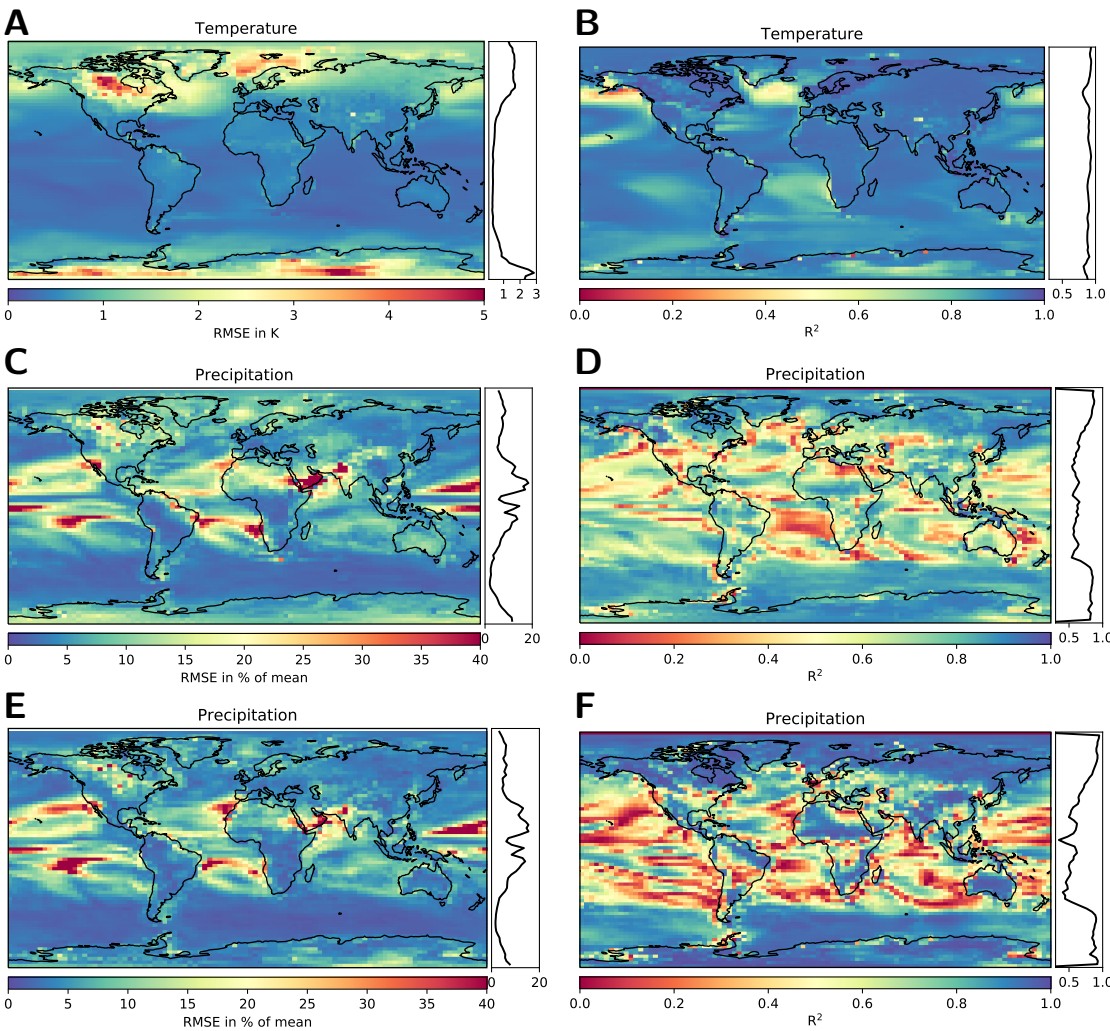

**Figure 4.** Left panel (A, C, E): Root mean square errors (RMSE) as estimators of the goodness of fit (lower is better) calculated using the test data. Right panel (B, D, F): $R^2$ values as estimator for the goodness of the model (higher is better) using the training data. Shown are the $R^2$ and RMSEs for (A,B) mean annual temperature, (C, D) precipitation, and (E, F) the alternative model for precipitation—based on temperature and specific humidity.



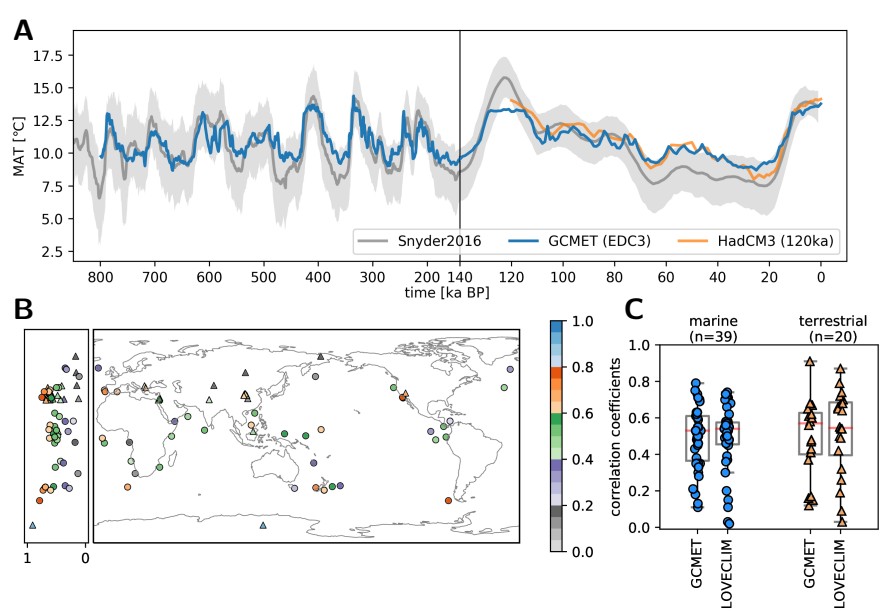

**Figure 5.** (*A*) Global mean temperature for the last 800 ka as predicted by GCMET based on different $CO_2$ records in comparison with a proxy-based global mean temperature reconstruction (Snyder, 2016). Furthermore, the time series from the 72 HadCM3 snapshots for the last 120 ka have also been added. Note the change in the spacing of the time axis at 140 ka BP. (*B*) Map of correlation coefficients between marine (in terms of as sea surface temperature) and terrestrial climate proxy time series and mean annual temperatures as reconstructed by GCMET-LO for the respective locations. The left panel shows the latitudinal distribution of the correlation coefficients. (*C*) Box plot of the correlation coefficients of GCMET and LOVECLIM with marine and terrestrial proxies.



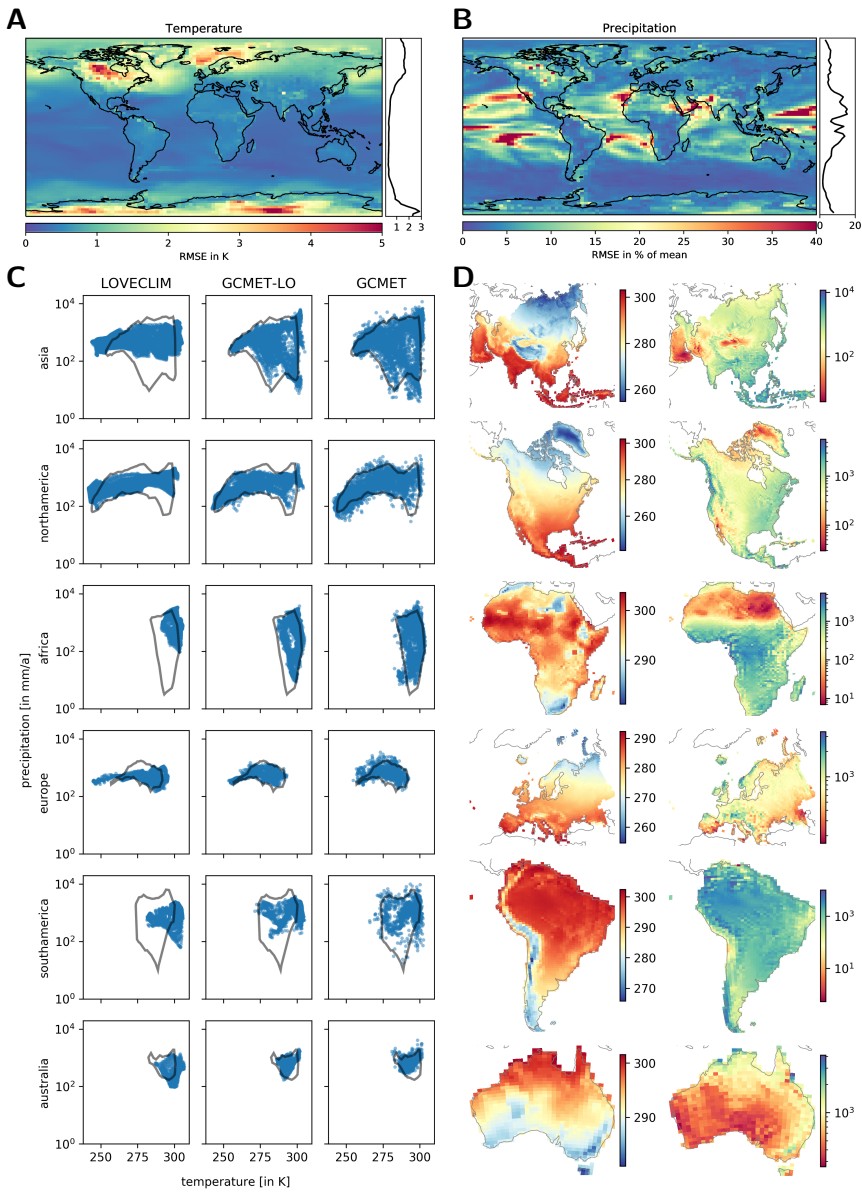

**Figure 6.** Root mean square error of the GCMET-LO predictions for the 14 HadCM3 snapshots for (*A*) MAT and (*B*) MAP (lower is better). (*C*) Present-day, i.e., 0 ka BP, temperature–precipitation phase diagram for Asia, North America, Africa, Europe, South America, and Australia, as modelled by LOVECLIM and reconstructed by GCMET-LO and GCMET and compared to observed multi-annual mean values (grey contours) for the period from 1961-1990 of the ERA-20C re-analysis data set (Poli et al., 2016). All model outputs have been bi-linearly interpolated onto the same grid, i.e. of the observational data, ERA-20C. (*D*) Maps of present-day temperature (in K) and precipitation (in mm/a) as reconstructed by GCMET for the six continents.







**Figure 7.** (A) Map of the 39 Middle and Late Pleistocene marine sea surface temperature proxies used in this study and tehir respective time series (B–E). Black dots indicate proxy sea surface temperature while blue lines indicate mean annual temperature as reconstructed by GCMET. Proxy–derived and model temperature are on the same scale, in °C.



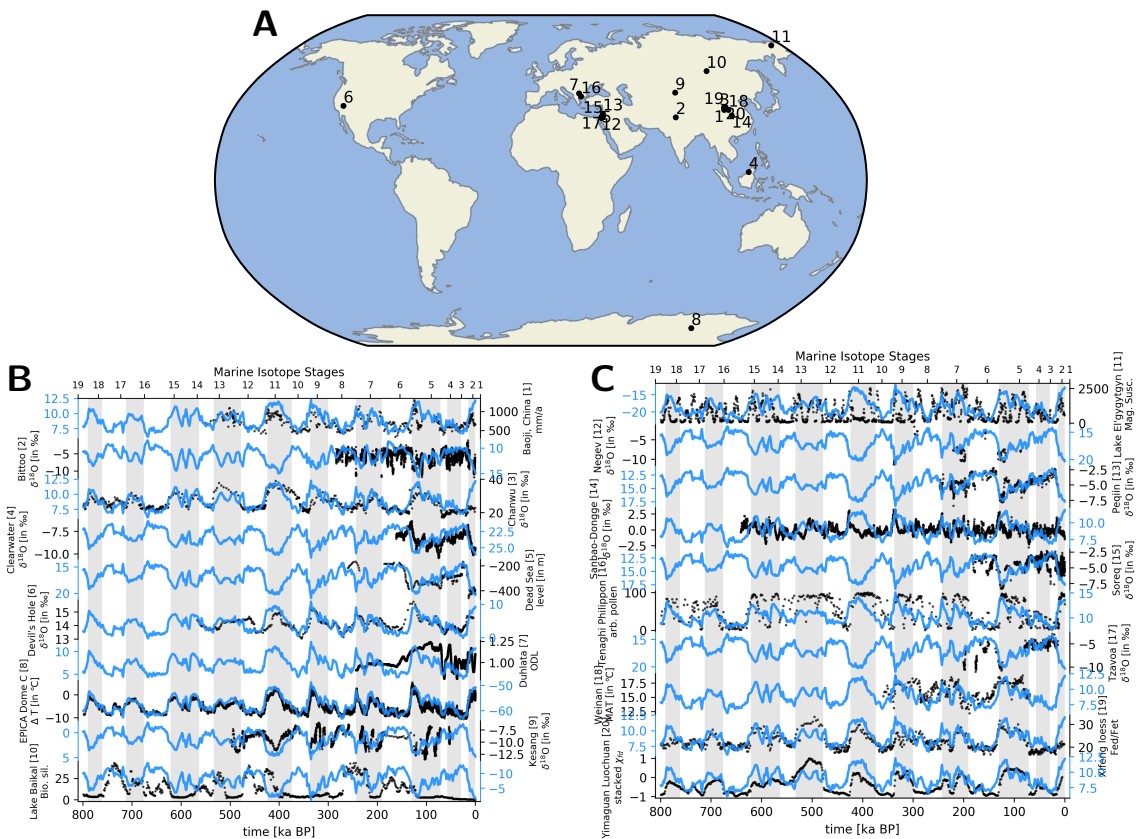

**Figure 8.** (A) Map of the 39 Middle and Late Pleistocene terrestrial climate proxies used in this study and their respective time series (B, C). Black dots indicate proxy variables (in different units) while blue lines indicate mean annual temperature as reconstructed by GCMET (in °C).





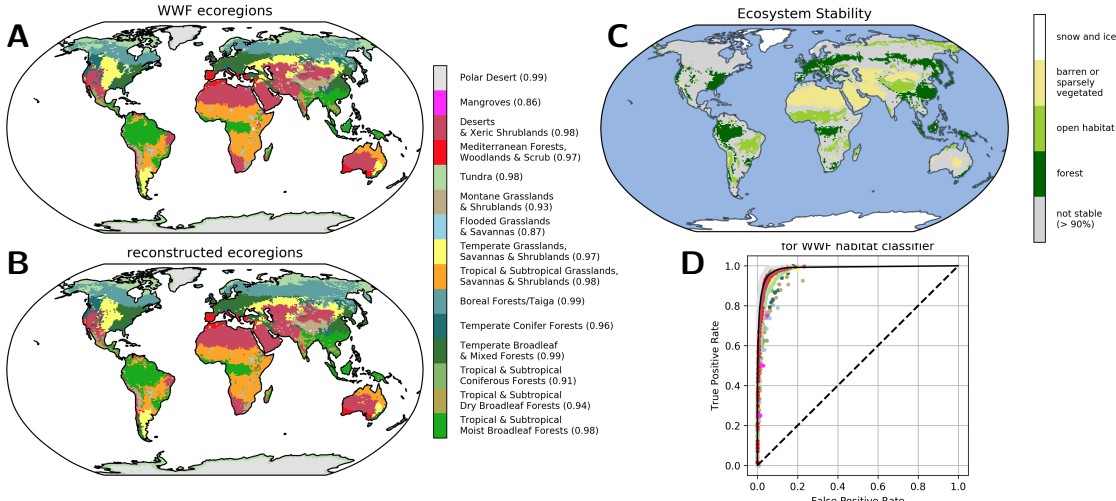

**Figure 9.** (*A*) Map of 14 major terrestrial habitats as defined by the WWF (Olson et al., 2001) for present-day and (*B*) as reconstructed with GCMET inputs of minimum and annual temperature and minimum and annual precipitation. (*C*) Stability of open habitats, such as grasslands and savannahs, and forest habitats, and sparsely vegetated regions across the world through the last 800,000 years. Regions in which the habitats have been unstable, i.e., of different type, for more than 90% are coloured in grey. (D) Receiver operating characteristic curve for the random forest classifier of the WWF 14 major habitats. The upper left corner represents a perfect prediction of an ecosystem, while the diagonal line represents a prediction made by random guessing. The closer the ROC curve is to the perfection point (0,1) the better the random forest classification is.