# Peer review of "A comprehensive history of climate and habitat stability of the last 800,000 years"

_Climate of the Past, 2019_

## Referee Comment (RC1) · Chris Brierley (Referee) · 13 Sep 2019

**Chris Brierley (Referee)**

c.brierley@ucl.ac.uk

Received and published: 13 September 2019

This paper presents a method for estimating climate and ecological changes over the past 800,000 years. This is incredibly ambitious and I welcome the endeavour and aims of the research. However, I am somewhat perturbed by its execution and cannot presently recommend it for publication. I have some queries that arose from my reading of the work. Some may be due to my own misunderstandings, but as a collection I feel they bring the validity of the proposed method into question.

The remit of this research is to build an emulator of climate and ecology as simulated by the HadCM3 model. Any errors in HadCM3 will therefore be unavoidably replicated by the emulator: yet this is not an obstacle preventing useful information being gleaned

from such a tool.

HadCM3 has previously been used to build emulators - and therefore several training sets exist (e.g. Arayo-Melo et al, 2015, and Lord et al., 2017). Given these have been designed to sample parameter space in a near-optimal form, I was surprised that the current work uses only snapshots of conditions that have existed over the past 120,000 years. The authors provide no explanation for this choice, nor discuss its limitations.

A second issue with this training set, as well as others, is the ice-sheet extent. In previous work, ice volume has either been considered an input parameter (Arayo-melo et al., 2015) or emulated through a fixed adjustment (Lord et al., 2017). Ice-dynamics, and the substantial lags that they introduce into the Earth system, are completely neglected in this work. This effectively assumes that ice sheet impacts are wholly and instantaneously determined by  $CO_2$  and orbital configuration. I anticipate this would explain elements of the model-data mismatch shown in Figs 6 7. There is also no recognition that last glacail cycle may not represent all glacial cycles (despite the mid-Brunhes transition).

A further unanswered question arising from the choice of training data revolves around the ecological reconstruction. Only surface temperature, humidity and precipitation are emulated, and then the biomes estimated off the back of this data. HadCM3 has an dynamic vegetation model (Triffid), although I'm unsure whether it was incorporated in this simulations. Certainly offline simulations of the Sheffield Dynamic Global Vegetation Model using the full HadCM3 climate model output have been performed for a subset of the training simulations (Singarayer et al., 2011, doi:10.1038/nature09739). Using these data could provide a useful comparison to the ecological modelling component in section 5 - i.e. can the emulator replicate the simulator response.

Moving beyond the training data choices, I have four questions about the choice of the method applied. Firstly, the linear (or log-linear) regression used at each grid point is different to previous efforts. Arayo-Melo et al. (2015) spent substantial effort developing
an approach that inherenetly builds in the spatail covariances inherent in climate using EOFs dimension reduction. The justification you give to avoiding this approach is that linear regression results in "well-behaved". Isn't this just another way of saying that you avoid non-linear transitions, but are these not a widely-accepted feature of the climate system. Also you should bear in mind that the whilst the functions are well-behaved in time, you have removed any such condition in space. Personally, I prefer the dimension reduction approach, as it pulls out climate features from any grid-point noise.

My second question about the methodology is why the function in equation 5 was selected for precipitation. Specific humidity is strongly related to temperature, unlike relative humidity) so it unclear to me that you can treat them as indepedent variables. The relationship between them can clearly be seen in Fig 2D - where the patterns are approximately opposite to each other.

My third methodological question revolves around downscaling. I appreciate your effort to downscale the climate results using High-resolution models. However, I wonder if you have applied them is the most optimal method. The (low-res) emulator captures climate changes from a (known) mean state. Your downscaling approach acts to modify those cliamte changes by modelling the resolution dependent aspects of those changes. If you want to convert the emulator output from climate anomalies to absolute climate, you must build back in the known mean state. Your choice of mean state is not explicit, and one wonders whether this might most appropriately be a very-high resolution satellite dataset (see question later about Fig. 6)

My final question about the methodology is where are the error bars on your estimate. Whilst I recognise that you cannot capture the error associated with HadCM3's biases, it must be possible to provide error bars of how well your model emulates the simulator. This is surely vital for the verification shown in Fig. 4 - does the true simulator response lie within the error bar estimates? What is the additional errors introduced by the simplicity of the ecosystem model applied to the emulator outputs. CPD
I have an additional question about the validation in Fig. 6 C and D. What is being assessing here? The emulator only models *changes* in climate, not absolute variables as shown in Fig. 6D. I suspect that the assessment in Fig. 6C is more about the resolution of underlying simulator and little to do with the emulator. In fact, previous efforts (e.g. Lord et al., 2017) have used the present-day climate as the "mean" from which anomalies are calculated - which instead potentially allows the use of ERA-20C as the baseline. Under the test shown in Fig. 6, such a slightly revised emulator would be perfect.

Given the quantity and importance of these questions about the creation and validation of the emulator, I have chosen not to review the results in any detail.

---

## Referee Comment (RC2) · Anonymous Referee #2 · 20 Sep 2019

This manuscript uses a set of snapshot simulations with HadCM3 over the last 120kyr to build an emulator that then extends into the last 800kyr. In addition a few atmosphere only simulations at higher spatial resolution than the coupled HadCM3 covering the more recent period are used to add more spatial variation into the long time-scale climate emulation.

While the approach is novel and seem very promising to me, I have still some difficulties to understand how well this approach is actually validatable. Maybe this is (partly) due to my limited understanding, but from reading the text and the figures it becomes not clear to me whether the approach is justified and how large potential errors could be. Therefore, in its present form I cannot recommend the paper for publication.

Fig 5 should be the most convincing figure to show that the emulated climate gains

quality because it uses – in addition to the HadCM3 snapshots covering the last 120kyr – additional information on spatial patterns from higher-resolution simulations. However, I get very confused by this figure and its discussion: Fig 5B shows GCMET-LO, while Fig. 5C shows LOVECLIM and GCMET. How can I see that GCMET (higher resolution I assume) does better in the spatial patterns than GCMET-LO and LOVECLIM?

Next, section 3 is meant to compare the emulated climate to proxy data. The overall (global) comparison is made in Fig 5A, while more detailed patterns are evaluated in Figs. 7 and 8. Fig 6 adds a (very useful) comparison to the present day climate by HadCM3. This whole section is again confusing. First of all, during the last 80kyr HadCM3 seems to be at the upper end of the data set (Snyder) in Fig. 5A, while in the reconstructed time period (before 140ka BP) blue and grey lines seem to match rather well. I don't see how we can account for a potential model bias (from the last 80kyr) in the earlier (emulated) period? Moreover, in the other figures it is unclear what is compared to LOVECLIM and what to GCMET-LO or GCMET? Why is the LOVECLIM simulation necessary here, as you also have a low resolution HadCM3 version?

Two more specific (but still general) remarks:

Line 153-54: Some patterns of climate variability most likely don't show up in the lower resolution snapshots. However, these might be very sensitive to orbital forcing and not only depend on just CO2. I don't see how you get a reliable addition to the low-resolution version by just adding the high-low resolution difference based on CO2 concentration.

Line 258: How important is it for the underlying snapshot simulations to be in 'quasi'-equilibrium? Can you estimate the error due to non-perfect equilibration of the training set?

---

## Author Comment (AC1) · 31 Oct 2019

We would like to thank Reviewer 1 for his comments on our discussion paper. Below, we have listed the reviewer's comments as quotes, our response in normal text, and action points in *italics*.

"This paper presents a method for estimating climate and ecological changes over the past 800,000 years. This is incredibly ambitious and I welcome the endeavour and aims of the research. However, I am somewhat perturbed by its execution and cannot presently recommend it for publication. I have some queries that arose from my reading of the work. Some may be due to my own misunderstandings, but as a collection

[Figure]

I feel they bring the validity of the proposed method into question.

The remit of this research is to build an emulator of climate and ecology as simulated by the HadCM3 model. Any errors in HadCM3 will therefore be unavoidably replicated by the emulator: yet this is not an obstacle preventing useful information being gleaned from such a tool.

HadCM3 has previously been used to build emulators - and therefore several training sets exist (e.g. Arayo-Melo et al, 2015, and Lord et al., 2017). Given these have been designed to sample parameter space in a near-optimal form, I was surprised that the current work uses only snapshots of conditions that have existed over the past 120,000 years. The authors provide no explanation for this choice, nor discuss its limitations."

**Our response**: It is true that previous emulators have been designed to sample parameter space that is near-optimal and that is as large as possible. We are limiting our approach to glacial–interglacial climates. Parameters outside that range are likely to lead to extrapolated climate reconstructions. Fortunately, the last 120ka have been extreme in terms of the glacial-interglacial climate ranges covered. The LGM ( 26ka BP) was one of the coldest periods and one with the largest glaciations of the Quaternary and likewise the last interglacial ( 115-130ka BP) was one of the warmest periods. In fact, the parameter sampling (despite being non-random) is much denser for glacial–interglacial climate emulation than any previous emulator designs. We use 72 time slices, more samples than have been used in previous studies, e.g., n=61 in Arayo-Melo et al. (2015) and n=40 in Lord et al. (2017). In summary, the sampling of the forcing is much closer to any forcing of the last 800ka compared to the previous emulator studies using HadCM3.

**ACTION**: *We will provide more details about our choice of the sampling strategy in the*

*introduction and the model description. We will highlight that our emulator approach uses more samples which themselves are much closer to the expected phase space trajectories, i.e., of the last 800ka. This way the emulator is only applicable to boundary conditions which are similar to the Quaternary ice age climates.*

"A second issue with this training set, as well as others, is the ice-sheet extent. In previous work, ice volume has either been considered an input parameter (Arayo-melo et al., 2015) or emulated through a fixed adjustment (Lord et al., 2017). Ice-dynamics, and the substantial lags that they introduce into the Earth system, are completely neglected in this work. This effectively assumes that ice sheet impacts are wholly and instantaneously determined by CO2 and orbital configuration. I anticipate this would explain elements of the model-data mismatch shown in Figs 6 7. There is also no recognition that last glacail cycle may not represent all glacial cycles (despite the mid- Brunhes transition)."

**Our response**: We know that our approach has limitations as outlined by the reviewer and indeed, ice-sheets are a factor accounting for the mismatch over ice-covered regions. However, we think this is more of a sampling problem rather than a representation of the temperature–ice sheet feedback. The HadCM3 model has been set up without interactive ice sheet models. As such the ice-sheets are fixed in size as well as in height and surface albedo. Therefore, we understand that the HadCM3 climate response is only being affected by the presence of ice sheets (and any dynamical circulation changes). So he neglect of ice dynamics is due to the HadCM3 setup and not so much due to the emulator itself. As a consequence, we have to accept "that ice sheet impacts are wholly and instantaneously determined by CO2 and orbital configuration", as correctly observed by the reviewer, as a limitation of the underlying HadCM3 simulations (but not the emulator itself).
Regarding the last point, we assume the emulator response to the last glacial cycle (i.e., the last 120ka) is representative of previous glaciations. We don't claim that all glaciations are equal. That is why we use a varying forcing (ice sheet masks, CO2, and orbital parameters) for all the previous glacial cycles for our reconstructions. Previous glacial cycles are therefore not simply repetitions of the last glacial cycle but linear combinations of the different responses to the external forcings. In this way, our emulator can be understood as a decomposition of the climate response into the various forcing components, e.g., $T = T_{orb} + T_{co2} + T_{mask}$.

**ACTION**: *: We will clarify the limitations of the HadCM3 setup early on in the model descriptions and we will add those raised issues in the discussion, i.e., what does this mean for the reconstructed climate.*

"A further unanswered question arising from the choice of training data revolves around the ecological reconstruction. Only surface temperature, humidity and precipitation are emulated, and then the biomes estimated off the back of this data. HadCM3 has an dynamic vegetation model (Triffid), although I'm unsure whether it was incorporated in this simulations. Certainly offline simulations of the Sheffield Dynamic Global Vegetation Model using the full HadCM3 climate model output have been performed for a subset of the training simulations (Singarayer et al., 2011, doi:10.1038/nature09739). Using these data could provide a useful comparison to the ecological modelling component in section 5 - i.e. can the emulator replicate the simulator response."

**Our response**: TRIFFID is part of this HadCM3 model setup, so this question makes perfect sense. Yes, we could use the vegetation model output, e.g., plant functional types, to compare the random forest classifier results with actual model output. Whereas we envisioned the ecological reconstruction as an example for how

such a dataset could be put to use, we now feel that it is more distracting from the main message of this paper. As the reviewer mentioned a comparison id required to assess the quality of the ecosystem reconstructions which would add another layer of complexity.

**ACTION**: *We would leave out the ecosystem reconstruction part in a revised paper because we feel that the reader may be overwhelmed with yet another reconstruction approach and its validation/evaluation. Instead, we would focus on the technical aspects of the emulator and provide a more comprehensive validation with HadCM3 model output.*

"Moving beyond the training data choices, I have four questions about the choice of the method applied. Firstly, the linear (or log-linear) regression used at each grid point is different to previous efforts. Arayo-Melo et al. (2015) spent substantial effort developing an approach that inherenetly builds in the spatail covariances inherent in climate using EOFs dimension reduction. The justification you give to avoiding this approach is that linear regression results in "well-behaved". Isn't this just another way of saying that you avoid non-linear transitions, but are these not a widely-accepted feature of the climate system. Also you should bear in mind that the whilst the functions are well-behaved in time, you have removed any such condition in space. Personally, I prefer the dimension reduction approach, as it pulls out climate features from any grid-point noise."

**Our response**: We claim that the emulator works perfectly fine for each grid point and that it recovers the climate response as in HadCM3. And yes, this implies that we don't control how grid points interact on different spatial scales. However, we tried to make the argument that the spatial coherence is recovered by the regression coefficients

themselves (see Fig. 2 in the discussion paper). To make this point clearer, we have attached plots of the covariance matrices for the (full) data set of both, HadCM3 and the emulator, GCMET (Figs. 1 & 2). The plots show the covariance between each grid point (96x73=7008) for the whole 72 time slices, i.e., a 7008x7008 covariance matrix, and the difference between the covariance matrix for both temperature and precipitation. The covariance matrix is a useful indicator for the representation of spatial structures in the emulator (and HadCM3). Smaller differences mean that GCMET captures the correct spatial covariances. As can be seen in the Figures, the covariance matrix for GCMET output is structurally similar to HadCM3 and the difference is mostly close to zero.

Regarding the non-linear transitions, we can only build on any assumptions that have been put into the HadCM3 simulations, or the climatological output. Although HadCM3 can account for non-linearities (it's a climate model after all), the climatological means (30-yr averages) on which we build GCMET, may not represent any non-linearities found in the climate system. We simply don't account for non-linear transitions because GCMET is supposed to be a surrogate model for the equilibrium climate model response.

A dimension reduction approach such as PCA removes lower-order variability modes in order to make the emulator approach computationally feasible whereas a linear regression model can make use of the full data set. In that sense, the linear regression is less restrictive about the inputs than a dimension reduction approach.

**ACTION**: *We will expand the discussion around different emulator approaches and point out advantages and disadvantages of using one approach over the other more thoroughly. Specifically, we will add an analysis to show that GCMET preserves spatial covariance sufficiently enough in comparison to the HadCM3 model outputs. A*

*preliminary figure is attached to this response.*

"My second question about the methodology is why the function in equation 5 was selected for precipitation. Specific humidity is strongly related to temperature, unlike relative humidity) so it unclear to me that you can treat them as indepedent variables. The relationship between them can clearly be seen in Fig 2D - where the patterns are approximately opposite to each other."

**Our response**: This is a valid point. In a strict sense, we shouldn't use specific humidity and temperature because of their collinearity. Admittedly, relative humidity is also a better proxy for precipitation: the closer the air mass is to saturation the more likely it is to rain. However, relative humidity is more difficult to predict.

**ACTION**: *We will provide the alternative formulation for precipitation based on relative humidity.*

"My third methodological question revolves around downscaling. I appreciate your effort to downscale the climate results using High-resolution models. However, I wonder if you have applied them is the most optimal method. The (low-res) emulator captures climate changes from a (known) mean state. Your downscaling approach acts to modify those cliamte changes by modelling the resolution dependent aspects of those changes. If you want to convert the emulator output from climate anomalies to absolute climate, you must build back in the known mean state. Your choice of mean state is not explicit, and one wonders whether this might most appropriately be a very-high resolution satellite dataset (see question later about Fig. 6)"

**Our response**: That is another very good point. Indeed, the mean state could be based on an observational data set (such as ERA-20C averages, Poli et al., 2016). This is a different way of saying that we need to add a bias correction to the emulator output. And we could. We thought that this is beyond the scope of our paper but we are happy to reconsider. In previous (discussion) paper we show that commonly-used bias correction methods already reduce model biases when comparing to paleo-climate observational data sets (Beyer et al., 2019). We can easily apply such a bias correction for our final data set. We want this paper also to be the reference for a comprehensive reconstruction of the past 800ka. So, it makes sense to have those data available as bias corrected emulator outputs.

**ACTION**: *We will bias-correct the emulator output and add a paragraph about the final climate reconstructions in the model description and the discussion section.*

> "My final question about the methodology is where are the error bars on your estimate. Whilst I recognise that you cannot capture the error associated with HadCM3's biases, it must be possible to provide error bars of how well your model emulates the simulator. This is surely vital for the verification shown in Fig. 4 - does the true simulator response lie within the error bar estimates? What is the additional errors introuded by the simplicity of the ecosystem model applied to the emulator outputs."

**Our response**: It is possible to estimate the uncertainty of the emulator in terms of a confidence interval (CI) with level confidence level $\alpha$, of a parameter p using the standard error (SE):

$$CI = p \pm q \cdot SE \quad with \quad q = CDF - 1(1 - \alpha/2)$$

As we mentioned before, we would drop the ecosystem reconstruction part for this

paper and, instead, focus on improving the model validation/verification part.

**ACTION**: *We will provide maps for the confidence interval (e.g., the 95% range) and (optionally) maps of the standard error. For Fig 5A, we propose to add the confidence levels so we can show how well the emulator results compare to HadCM3 and to the global mean temperature time series.*

"I have an additional question about the validation in Fig. 6 C and D. What is being assessing here? The emulator only models changes in climate, not absolute variables as shown in Fig. 6D. I suspect that the assessment in Fig. 6C is more about the resolution of underlying simulator and little to do with the emulator. In fact, previous efforts (e.g. Lord et al., 2017) have used the present-day climate as the "mean" from which anomalies are calculated - which instead potentially allows the use of ERA-20C as the baseline. Under the test shown in Fig. 6, such a slightly revised emulator would be perfect."

**Our response**: That is a good point and would fall within the proposed changes of our next to last comment.

**ACTION**: *see our next to last comment*

"Given the quantity and importance of these questions about the creation and validation of the emulator, I have chosen not to review the results in any detail."

**Our response**: We regard the comparison to proxy data as one of the essential pieces of the paper. This part would serve as the ultimate test showing how well our emulator

results compare to observations (any previous validation was purely restricted to the "model" world of HadCM3 and GCMET).

References

- Araya-Melo, P. A., Crucifix, M., and Bounceur, N.: Global sensitivity analysis of the Indian monsoon during the Pleistocene, Clim. Past, 11, 45-61, https://doi.org/10.5194/cp-11-45-2015, 2015.

- Lord, N. S., Crucifix, M., Lunt, D. J., Thorne, M. C., Bounceur, N., Dowsett, H., O'Brien, C. L., and Ridgwell, A.: Emula- tion of long-term changes in global climate: application to the late Pliocene and future, Climate of the Past, 13, 1539-1571, https://doi.org/10.5194/cp-13-1539-2017, 2017.

- Poli, P., Hersbach, H., Dee, D. P., Berrisford, P., Simmons, A. J., Vitart, F., Laloyaux, P., Tan, D. G. H., Peubey, C., Thepaut, J.-N., Tremolet, Y., Holm, E. V., Bonavita, M., Isaksen, L., and Fisher, M.: ERA-20C: An Atmospheric Reanalysis of the Twentieth Century, J. Climate, 29, 4083-4097, https://doi.org/10.1175/JCLI-D-15-0556.1, 2016.

- Singarayer, J. S. and Valdes, P. J.: High-latitude climate sensitivity to ice-sheet forcing over the last 120kyr, Quaternary Science Reviews, 29, 43-55, https://doi.org/10.1016/j.quascirev.2009.10.011, 00092, 2010.

[Figure]

[Figure]

**Fig. 1.** Spatial covariance matrices of temperature (in units of K2) calculated for all grid points (n = 96x73 = 7008) of HadCM3 (left), GCMET (middle), and the difference of the two covariance matrices (right

[Figure]

**Fig. 2.** Spatial covariance matrices of precipitation (in units (mm/a)2), calculated for all grid points (n = 96x73 = 7008) of HadCM3 (left), GCMET (middle), and the difference of the two covariance matrices (

---

## Author Comment (AC2) · 31 Oct 2019

We would like to thank Reviewer 2 for the constructive comments on our discussion paper. Below, we have listed the reviewer's comments as quotes, our response in normal text, and action points in *italics*.

"This manuscript uses a set of snapshot simulations with HadCM3 over the last 120kyr to build an emulator that then extends into the last 800kyr. In addition a few atmosphere only simulations at higher spatial resolution than the coupled HadCM3 covering the more recent period are used to add more spatial variation into the long time-scale climate emulation. While the approach is novel and seem very promising to me, I have still some diffi-

culties to understand how well this approach is actually validatable. Maybe
this is (partly) due to my limited understanding, but from reading the text
and the figures it becomes not clear to me whether the approach is justified
and how large potential errors could be. Therefore, in its present form I
cannot recommend the paper for publication.

   Fig 5 should be the most convincing figure to show that the emulated cli-
mate gains quality because it uses - in addition to the HadCM3 snapshots
covering the last 120kyr - additional information on spatial patterns from
higher-resolution simulations. However, I get very confused by this figure
and its discussion: Fig 5B shows GCMET-LO, while Fig. 5C shows LOVE-
CLIM and GCMET. How can I see that GCMET (higher resolution I assume)
does better in the spatial patterns than GCMET-LO and LOVECLIM?"

**Our response**: Apparently, there is a typo in the labels. Sorry about that. Fig. 5C
shows how GCMET-LO matches proxy reconstructions in terms of the correlation
coefficient compared to LOVECLIM model output. Fig. 5A shows how well GCMET
compares with proxy-based global mean temperature estimate throughout the last
800ka. Although it has a higher spatial resolution, this figure was not intended to
suggest that this is because of the higher resolution. The whole Fig. 5 is intended
to highlight the emulator's capability to reconstruct the temperature change signal
realistically enough, i.e., the temporal, climate change signal. Complementary, Fig.
6 is intended to show how much more spatial detail can be recovered by adding
the higher-resolution corrections on to of GCMET-LO, and how the different spatial
resolutions compare to the observed (i.e., present-day) ranges of temperature and
precipitation (from the ERA-20C re-analysis dataset, Poli et al., 2016). In conclusion,
the emulator recovers the observed climate change signal through time at different
(proxy) locations, and it also recovers the correct shape of the observed spatial climatic
heterogeneity across the different continents.

**ACTION**: *This part of the paper seems to be confusing. Therefore, we would restructure Sections 2 and 3 to clarify that we are showing two different, but unrelated, aspects of the emulator: i) the temporal component, covered by the coarse-resolution HadCM3 emulation and its comparison (currently Section 3), and ii) the spatial component, covered by the dynamical downscaling (currently Section 2.3). The dynamical downscaling section would then follow after the proxy-comparison.*

"Next, section 3 is meant to compare the emulated climate to proxy data. The overall (global) comparison is made in Fig 5A, while more detailed patterns are evaluated in Figs. 7 and 8. Fig 6 adds a (very useful) comparison to the present day climate by HadCM3. This whole section is again confusing. First of all, during the last 80kyr HadCM3 seems to be at the upper end of the data set (Snyder) in Fig. 5A, while in the reconstructed time period (before 140ka BP) blue and grey lines seem to match rather well. I don't see how we can account for a potential model bias (from the last 80kyr) in the earlier (emulated) period? Moreover, in the other figures it is unclear what is compared to LOVECLIM and what to GCMET-LO or GCMET? Why is the LOVECLIM simulation necessary here, as you also have a low resolution HadCM3 version?"

**Our response**: The mismatch between HadCM3 output and the global mean temperature (Snyder, 2016) can be explained by two sources of error:

1. GCM model error/bias: this is due to the representation of the climate system and dynamics which is intrinsic to the model. We can't do anything about this type of error.

2. Errors in the boundary conditions: The HadCM3 boundary conditions, such as $CO_2$ or ice sheet configuration, are not the latest, i.e., best estimates of boundary conditions. $CO_2$, for example, has been kept as in the original study by

Singarayer and Valdes, 2010, which is based on the Vostok ice core record. In the meantime, the best CO2 reconstruction for the past comes from the EPICA Dome C ice core (Bereiter et al., 2015). Likewise, the ice sheet extent is based on ICE-5G (Peltier, 2004) which is now superseded by ICE-6G (Peltier et al., 2015). We can do something about this type of error. Because GCMET represents the response of HadCM3 to external forcings, we can use corrected versions thereof. In our 800ka reconstructions, we use CO2 from the EPICA Dome C record, and ice sheet reconstructions based on Ganopolski et al. (2011), ICE-6G covers only the last 122ka.

In general, the flexibility of an emulator approach allows to explore different boundary conditions with their associated uncertainty. Such an exploration is not feasible with state-of-the-art GCMs/EMICs.

The comparison with LOVECLIM is necessary because it is one of very few model results which extend that far back in time in a continuous way. The cross-comparison between GCMET-LO and LOVECLIM allows the reader see that the emulator possesses reasonable skill to model/reconstruct the climate. Again, it seems that most of the confusion is avoidable if we restrict our technical analysis to the aforementioned two aspects of i) temporal climate change signal and ii) the spatial detail which comes along with the dynamical downscaling. The basic idea for bringing those two aspects together is that we can provide the best paleo-climate dataset for the last 800ka, that is both reasonable in time and in space, for the wider paleo-modelling/application community.

**ACTION**: *We will make our intent clearer: that we want to deliver the best and most comprehensive climate reconstructions for the last 800ka. To reduce the confusion of why we compare different models and different resolutions, we would simply drop*

*the comparison with LOVECLIM in the main text and move this cross-comparison part into the supplementary of the paper. For the comparison to global mean temperature (Fig. 5A), we will additionally provide the lower and upper confidence interval of our reconstructions to give the reader an idea about the emulator uncertainty with respect to global mean temperature. The flexibility of the emulator to quickly explore different boundary conditions will be discussed in the discussion section as one of the strengths of such an approach. We will argue that imperfect boundary conditions (such as older $CO_2$ estimates) during the time of the GCM simulations can be corrected afterwards in a modified emulator re-run.*

"Two more specific (but still general) remarks:

Line 153-54: Some patterns of climate variability most likely don't show up in the lower resolution snapshots. However, these might be very sensitive to orbital forcing and not only depend on just CO2. I don't see how you get a reliable addition to the low-resolution version by just adding the high-low resolution difference based on CO2 concentration."

**Our response**: That is a good point. Unfortunately, we currently have only 9 high-resolution data points and with too few samples we can't use more than one regressor. We can try to use a second regressor (a rule of thumb is: more than 3*k sample points for k regressors, which means we could use 2 regressors), or alternatively define the first two principal components of all the forcings, CO2 and orbital parameters. This would require further analysis. However, because the spatial downscaling happens after the emulator reconstruction (provided at HadCM3-resolution), we could leave the spatial, dynamical downscaling part for another paper, once we've solved this issue sufficiently. This means that the results shown in Fig. 6C and D would be removed from the paper. Fig. 9 would be updated, depending on which high-resolution maps we generate from the bias-correction.

**ACTION**: *We will update our analysis based on a higher resolution mean state, i.e., an observational data set (such as ERA-20C, Poli et al, 2016). This dataset will also be used for the bias correction of our climate reconstructions which we want to make publicly available.*

"Line 258: How important is it for the underlying snapshot simulations to be in 'quasi'- equilibrium? Can you estimate the error due to non-perfect equilibration of the training set?"

**Our response**: If the snapshot simulations are in (quasi-)equilibrium then we can assume that the climate system response that we capture with GCMET is the equilibrium response. However, if the climate simulations are not in equilibrium than it is possible that the transient response is non-linear. Unfortunately, there is no clear definition of "the climate model being in equilibrium". That means that we can't quantify how far the HadCM3 simulations are away from that equilibrium. Usually, GCMs tend to drift, specifically due to the slow dynamics of the deep ocean. It can take several thousands of simulation years to reach equilibrium, so we can only assume that HadCM3 is sufficiently equilibrated. Here is a quote from Singarayer and Valdes (2010) that summarizes this issue: *"Although not in perfect equilibrium (most GCMs are never truly in equilibrium), we judged that we were close enough for the models to be representative of the time period [...]. The 500 year length of integration was typical of many models used within the PMIP2 project"*. Without a fully transient GCM simulation over a sufficiently long period, from which we can infer the difference between transient and quasi-equilibrium snapshot, it is hard to make a judgment about the importance of this issue.

**ACTION**: *We will discuss this, as outlined above, in more depth in the discussion*

*section of the paper.*

**References**

- Bereiter, B., Eggleston, S., Schmitt, J., Nehrbass-Ahles, C., Stocker, T. F., Fischer, H., Kipfstuhl, S., and Chappellaz, J.: Re- vision of the EPICA Dome C CO2 record from 800 to 600 kyr before present, Geophys. Res. Lett., 42, 2014GL061957, https://doi.org/10.1002/2014GL061957, 2015.

- Ganopolski, A. and Calov, R.: The role of orbital forcing, carbon dioxide and regolith in 100 kyr glacial cycles, Clim. Past, 7, 1415-1425, https://doi.org/10.5194/cp-7-1415-2011, 2011.

- Peltier, W. R. Global glacial isostasy and the surface of the ice age Earth: the ICE-5G (VM2) model and GRACE. Annu. Rev. Earth Planet. Sci. 32, 111-149 (2004)

- Peltier, W.R., Argus, D.F. and Drummond, R. (2015) Space geodesy constrains ice-age terminal deglaciation: The global ICE-6G_C (VM5a) model. J. Geophys. Res. Solid Earth, 120, 450-487, doi:10.1002/2014JB011176.

- Poli, P., Hersbach, H., Dee, D. P., Berrisford, P., Simmons, A. J., Vitart, F., Laloyaux, P., Tan, D. G. H., Peubey, C., Thepaut, J.-N., Tremolet, Y., Holm, E. V., Bonavita, M., Isaksen, L., and Fisher, M.: ERA-20C: An Atmospheric Reanalysis of the Twentieth Century, J. Climate, 29, 4083-4097, https://doi.org/10.1175/JCLI-D-15-0556.1, 2016.

- Singarayer, J. S. and Valdes, P. J.: High-latitude climate sensitivity to ice-sheet forcing over the last 120kyr, Quaternary Science Reviews, 29, 43-55, https://doi.org/10.1016/j.quascirev.2009.10.011, 00092, 2010.

- Snyder, C. W. (2016). Evolution of global temperature over the past two million years. Nature, 538(7624), 226.